# Transvenous Lead Extraction in a European Low-Volume Center without On-Site Surgical Support

**DOI:** 10.3390/reports6030041

**Published:** 2023-09-07

**Authors:** Mohamed Dardari, Corneliu Iorgulescu, Vlad Bataila, Alexandru Deaconu, Eliza Cinteza, Radu Vatasescu, Paul Padovani, Corina Maria Vasile, Maria Dorobantu

**Affiliations:** 1Department of Cardiology, Carol Davila University of Medicine and Pharmacy, 050474 Bucharest, Romania; dardarimohamed@yahoo.com (M.D.); vladbataila@yahoo.co.uk (V.B.); alexandru.deaconu@umfcd.ro (A.D.); maria.dorobantu@umfcd.ro (M.D.); 2Electrophysiology and Cardiac Pacing Lab, Clinical Emergency Hospital, 014461 Bucharest, Romania; iorgulescu_corneliu@yahoo.com; 3Department of Pediatrics, Carol Davila University of Medicine and Pharmacy, 050474 Bucharest, Romania; 4Pediatric Cardiology Department, Marie Sklodowska Curie Children Emergency Hospital, 041451 Bucharest, Romania; 5Department of Pediatric Cardiology and Pediatric Cardiac Surgery, Nantes Université, CHU Nantes, FHU PRECICARE, F-44000 Nantes, France; padovanipaul17@gmail.com; 6Pediatric and Adult Congenital Cardiology Department, M3C National Reference Centre, Bordeaux University Hospital, F-33000 Bordeaux, France; corina.vasile93@gmail.com

**Keywords:** device-related endocarditis, transvenous lead extraction, cardiac implantable electronic device, pacemaker, defibrillator, pocket infection, non-powered lead extraction sheaths

## Abstract

Indications for cardiac implantable electronic devices (CIEDs) are increasing. Almost one-third of device-related infections are endocarditis. Transvenous lead extraction (TLE) has emerged as an effective and safe approach for treating device-related infections and complications. Multiple types of extraction tools are being used worldwide. Our goal is to evaluate the safety and effectiveness of TLE using non-powered extraction tools. The study included patients between October 2018 and July 2022 requiring TLE according to EHRA expert consensus recommendations on lead extraction. A total of 88 consecutive patients were included. Indications for TLE included device-related infections in 74% of the patients. Of those, 32% had device-related endocarditis with or without sepsis. Staphylococcus Aureus was the most frequent pathogen in patients with endocarditis and positive bacteremia, and 57% had negative bloodstream cultures. A total of 150 cardiac pacing and defibrillator leads were targeted for extraction. The mean dwell time for leads was 6.92 ± 4.4 years; 52.8% were older than 5 years, 15.8% were older than ten years, and the longest lead dwell time was 26 years. Patients’ age varied between 18 and 98, with a mean age of 66 ± 16 years. Sixty-seven percent of patients were males. Using only non-powered extraction tools, we report 93.3% complete lead removal and 99% clinical success with partial extraction. We report no procedure-related death nor major complications. Minor complication incidence was 6.8%, and all complications resolved spontaneously. The 30-day mortality rate was 3.4%. TLE using non-powered extraction tools is safe and effective even without surgical backup on site.

## 1. Introduction

Cardiac implantable electronic devices (CIED) are being increasingly used. Reports estimate that over one million CIEDs are implanted worldwide annually, with over 4.5 million active devices [1,2,3]. Complications subsequently have increased in parallel [4,5,6,7,8,9,10,11,12,13]. With a longer patient life expectancy, lead-revision and lead extraction procedures rapidly increase for device-related complications, including local or systemic infection system upgrades and lead malfunction [5,6,7,8,9,10,11,12,14,15,16]. The current annual trans-venous lead extraction (TLE) rate is estimated to have reached 10,000 to 15,000 procedures worldwide [17]. Indications for TLE have been defined in the 2018 EHRA expert consensus statement on lead extraction and the 2017 HRS expert consensus statement on lead management and extraction [18,19]. The incidence of TLE procedure-related complications has been reported to vary between 0.19 and 6.2%. Major complications include cardiac avulsion, vascular laceration, massive pericardial effusion, and death. Minor complications include pocket hematoma requiring evacuation, pericardial effusion, and pulmonary embolism without further intervention. Success rates reported in the literature vary between 93% and 98% using various extraction tools and techniques [19,20]. Few papers have addressed the efficacy and safety of TLE [21,22,23].

The “ELECTRa” registry is a recent, large prospective, multicenter, European controlled registry of consecutive patients undergoing total lead extraction procedures in European countries, including long-term follow-up data which included more than 3500 patients from 100 centers across Europe, emphasizing the importance of this issue [24]. The most common indication for TLE (accounting for more than 50% of the cases) is device-related infection. The only effective treatment of infected CIEDs is completely removing the infected hardware and capsulectomy, followed by individualized antimicrobial therapy [25,26,27,28,29,30,31]. Several methods for lead extraction have been reported to achieve high success rates and few complications, such as laser sheaths [32,33,34] or radiofrequency devices [35,36]. However, these methods’ availability is limited in most centers. The superiority of powered sheaths (laser, radiofrequency, etc.) over non-powered tools has been questioned in several papers in terms of success rates and cost-effectiveness [37,38].

The non-powered technique with multiple venous entry sites has been used and described by Bongiorni et al. to remove over 2000 pacing and implantable cardiac defibrillator (ICD) leads with a very high success rate and very low complication rates [38]. This technique requires an experienced operator to achieve the reported favorable outcome, and the reproducibility of this method in low-volume centers may be questioned. Further data are needed to evaluate the possibility of safely and effectively performing TLE using non-powered tools on a large scale, including low-volume centers without surgical backup on site. We have adopted Bongiorni’s technique in our center. Our goal was to evaluate the safety and effectiveness of this technique with no on-site surgical backup in a low-volume center (<30 procedures per year).

## 2. Materials and Methods

### 2.1. Study Type

This retrospective study was performed at the Clinical Emergency Hospital of Bucharest, Romania, a referral center for transvenous lead extraction. All patients referred to our center who fulfilled all the criteria for lead extraction procedures were included between October 2018 and July 2022. The LED score was not calculated as we believe it would not be accurate to apply in a low-volume center, where the difficulty can be overestimated if related to fluoroscopy time.

### 2.2. Study Participants

No exclusion criteria were applied other than the lack of patient consent for the procedure. Demographic and patient characteristics data and procedural data were noted, including the different types of approaches and extraction tools. Device data were collected, such as device and lead type, lead dwell time, number of leads, the type of lead fixation, and the location of leads. The outcome was noted and analyzed.

The study was conducted in accordance with the Declaration of Helsinki and approved by the Ethics Committee of Clinical Emergency Hospital of Bucharest, decision number 42311/24 October 2022. Informed consent was obtained from all patients after a detailed explanation of risks and outcomes.

### 2.3. Definitions

Lead Extraction Procedure: Our study considered a lead extraction procedure when extracting a lead older than 1 year or requiring specialized extraction tools, regardless of its duration or the need for a new venous route.

Success: Success was defined as achieving the desired outcomes related to the goal of the procedure. Complete success was defined when all leads were completely removed or when the targeted leads were successfully removed while leaving in place the rest of the leads that were not subject to the lead extraction procedure. Complete success was further contingent on the absence of death or permanent disability resulting from the procedure.

Infectious Complications: Infectious complications were defined according to the infectious definitions outlined in the 2018 EHRA expert consensus statement. Cardiac device-related endocarditis refers to the presence of lead or cardiac vegetation(s) detected through transesophageal echocardiography (TOE) and positive blood culture indicating infection without any evident source other than the cardiac implantable electronic device (CIED). Device pocket infection was determined based on signs of inflammation in the pocket area, such as local inflammation, skin necrosis, purulent discharge through a fistula, or any protrusion of the device from the skin. All major and minor complications were noted.

Diagnostic Management: In cases where bacteremia was detected without an apparent source other than the CIED and vegetation was not visible in TOE, additional diagnostic measures such as PET/CT with FDG were not pursued, and subsequently, TLE was performed.

Partial Success: Partial success was observed in cases where all leads were extracted. However, residual material within less than 4 cm from one of the leads could not be extracted transvenously. However, this residual material did not hinder the intended goals of the procedure, such as infection resolution or the creation of intravascular space, nor did it increase the risk of procedure-related complications.

Failure: Failure was defined as the inability to achieve procedural success, including extracting all leads to cure an infection or removing the targeted lead(s) for non-infectious complications. Any procedure-related death or permanently disabling complications were also considered a failure.

### 2.4. TLE Indication

The most common indications for TLE was a device-related infection (cardiac device related-endocarditis or pocket infection), followed by venous occlusion and other indications (dysfunctional leads or abandoned leads)

Infectious indications for lead extraction

Several types of CIED-related infections have been defined. Usually, superficial incisional infections resolve spontaneously or with minimal medical treatment and local disinfection with iodine solution and do not require TLE; however, for the other infection types, whether they present as local or systemic infections, including endocarditis, the entire device and the leads must be removed to cure the infection and limit the consequences.

Pocket infection

Local pocket infection is usually limited to the generator. It presents with local clinical signs of inflammation such as erythema with or without swelling of the pocket, tenderness and warmth of the skin and/or fluctuance (Figure 1), wound dehiscence, and/or purulent drainage. In this type of CIED-related infection, leads are intact and blood cultures are negative. This indication represented 56.9% of all infections requiring TLE in our study.

Device-related infections may not always be accompanied by the local signs of inflammation described above, as one type of pocket infection presents with skin erosion and/or device exposure without any other signs or symptoms of inflammation (Figure 2 and Figure 3). Patients may only describe local pain when presented with this condition; however, in this case, the device should always be considered infected, and the system should be completely removed.

Systemic infection and/or endocarditis

CIED endocarditis is challenging to diagnose and usually requires positive blood cultures with or without the documentation of lead or valvular vegetations. It is a life-threatening complication representing 10% of all cases of endocarditis [39]. Infection may primarily involve the pocket after direct handling (e.g., changing the generator). It may disseminate to the lead, producing multiple vegetations, or originate directly from the leads during bacteremia secondary to a minor infection outbreak. In addition to the typical risk factors for infective endocarditis (renal failure, corticosteroid use, congestive HF, and diabetes mellitus), other factors related to the surgical procedure may play a role in infective endocarditis-CIED (e.g., the type of implanted device, device revision, the use of temporary pacing, the use of antimicrobial prophylaxis, and the use of anticoagulation). The Modified Duke Criteria [40], used to diagnose valvular endocarditis, is frequently used along with additional lead-related criteria for the positive diagnosis of device-related endocarditis. Lead-related criteria for a positive diagnosis of endocarditis are (a) positive cultures of the extracted lead even in the absence of positive blood cultures, (b) the presence of lead vegetations documented using various imaging techniques such as transthoracic echocardiography (TTE) and TOE, and (c) abnormal metabolic activity around the CIED generator and/or leads detected via FDG PET/CT or radiolabeled leucocyte single-photon-emission computed tomography/CT. In the setting of bacteremia without an obvious source and in the presence of a CIED, the presumption should be CIED-related systemic infection, especially if bacteremia disappears after complete system removal. In our study, endocarditis and/or sepsis accounted for 43.1% of CIED-related infections referred to our hospital for TLE, and the most frequently detected pathogen was Staphylococcus Aureus.

Other indications for lead extraction

The ELECTRa registry described lead dysfunction as the second most frequent reason for TLE, accounting for 38.1% of the cases [24].

### 2.5. Statistical Analysis

All data collected were introduced in a structured database and were analyzed using IBM SPSS Statistics software V 26.0.

Patient information was collected and organized in an electronic database using GraphPad Prism Demo 6.0 software (GraphPad, San Diego, CA, USA) for data analysis. The Microsoft Excel statistics software suite (Microsoft, Albuquerque, NM, USA) was also employed in parallel for data management purposes.

Descriptive statistics and histograms were employed to examine the distribution of the data. Statistical analysis was performed using appropriate methods, including Student’s *t*-test, the chi-square test, and the Pearson correlation coefficient, as deemed suitable for the analysis.

### 2.6. TLE Procedure

All procedures in the electrophysiology laboratory were conducted by two physicians and two nurses. Although surgical backup was available on call, no surgeons were available on site. Before the procedures, patients were fasting, and a thorough pre-procedural evaluation was performed for all individuals undergoing transvenous device removal. This evaluation included noting the type of device and lead and the age of the implant. Chest X-rays were taken at various angles to visualize the intravascular route of the leads and identify any signs of lead damage or insulation rupture.

Device interrogation was conducted to assess lead function parameters, such as lead impedance, pacing threshold, and the patient’s underlying rhythm. In cases of device-related infections, inflammation markers and blood cultures were obtained. Transthoracic echocardiography (TTE) was utilized to determine the positioning of the lead tip (e.g., right ventricle free wall or intra-ventricular septum), detect the presence of cardiac vegetations, assess the association between vegetations and the leads, and gather data on the patient’s cardiac function. Transesophageal echocardiography (TOE) was performed selectively when deemed necessary.

For patients with device infections, broad-spectrum antibiotic therapy was initiated before the procedure to minimize the extent of the infection. These measures ensured comprehensive pre-procedural assessment and appropriate management for patients undergoing transvenous lead extraction in the electrophysiology laboratory.

In most procedures, local anesthesia was administered using a 1% lidocaine solution, except for three cases requiring general anesthesia. A continuous electrocardiogram (ECG) and non-invasive blood pressure monitoring were implemented throughout the procedures. A temporary pacing lead was placed in the right ventricle (RV) to ensure backup pacing. Central venous system access was established using a 6-French femoral sheath, which remained on standby. Pericardiocentesis kits were readily available on site, and cardiac surgery backup was accessible on call.

Bipolar electrocautery was used for capsulectomy to detach the device and leads from muscular adhesions in the pocket, reducing bleeding. In cases where oral anticoagulation was ongoing, it was ceased 48 h before the procedure. Patients receiving vitamin K antagonists (VKA) underwent the operation only after their International Normalized Ratio (INR) had returned to normal. Bridging with low-molecular-weight heparin (LMWH) was rarely necessary. All-cause mortality was assessed during a 30-day follow-up period following the procedure.

For non-infectious indications, the targeted leads included abandoned or dysfunctional leads and leads that needed to be extracted to create intravascular space for the implantation of new leads. Other leads were left in place. All leads, including the device, were targeted for extraction in cases involving infectious indications.

For lead extraction, a minimally invasive approach was performed. Initially, minimal simple traction was applied through the venous route that was used for lead implantation. If this traction was insufficient, dedicated non-powered extraction tools, such as the Byrd Dilator Sheath Telescoping Polypropylene and Needle Eye Snare from Cook Medical (Cook Medical, Bloomington, IN, USA), were utilized. The technique involved ligating and applying continuous gentle traction to the extravascular end of the targeted lead using a surgical suture thread to maintain constant tension.

Subsequently, the Byrd dilator sheath was employed via rotating and advancing it around the targeted indwelled lead along its length and tip. While the lead remained under tension, the surrounding cardiac tissue would rupture, freeing the lead from intravascular and intracardiac adhesions. In cases where the mechanical traction and rotation of the sheaths caused lead fracture or rupture, rendering the remaining lead portion unreachable from the initial venous route (subclavian vein), an alternative venous route (jugular or femoral) was used. The needle eye snare was introduced through the alternative route up to the fractured fragment, allowing traction and extraction.

It is important to note that no powered mechanical tools or laser sheets were utilized during the lead extraction procedures, emphasizing the reliance on non-powered extraction tools and techniques.

Following the procedure, an echocardiography was performed on all patients in the electrophysiology lab. Subsequently, patients were closely monitored in the intensive care unit (ICU) for 24 h. In cases of pocket infection, targeted or empiric antimicrobial therapy was continued for a minimum of 7 days. For patients with systemic infection and/or infective endocarditis, antimicrobial therapy was extended to at least 14 days.

To evaluate the progression of local infections, systemic inflammation markers and clinical evolution were assessed daily. Daily cardiac echography, including transesophageal echocardiography where necessary, and repeated blood cultures were performed to monitor the evolution of patients with systemic infection.

These comprehensive monitoring measures aimed to track the response to antimicrobial therapy, assess the resolution of local and systemic infections, and evaluate the overall clinical progress of the patients following the lead extraction procedure.

## 3. Results

A total of 88 patients were included in the analysis, with 74% of transvenous lead extraction (TLE) procedures performed for some type of infection, while 26% were for venous occlusion or abandoned/dysfunctional leads.

The targeted leads for achieving procedural success amounted to n = 150, and an impressive 99.3% of these targeted leads were successfully extracted. Among the extracted leads, 93.3% were completely removed without any residual material. Simple traction alone was proved to be effective in only a small percentage (11.5%) of the leads.

In nine patients, traction with a snare device through the femoral route was combined with transvenous mechanical dissection for complete removal.

In three cases, successful extraction required sheath rotational dissection through the jugular route. A snare device via a right-femoral approach was preferred for one patient with a complete intravascular lead. Partial extraction of leads was observed in 6% of cases, but clinical success was achieved as the primary goal of TLE.

One reported failure (1.13% of patients) necessitated referral to cardiac surgery for surgical tricuspid valve replacement due to severe symptomatic regurgitation.

Minor procedure-related complications were observed in 6.8% of patients, including three cases of ventricular arrhythmias or conduction disorders, two cases of local pocket hematoma, and one pericardial effusion that did not require medical intervention. Notably, no major complications or procedural deaths were reported in the study.

Our study cohort’s 30-day mortality rate was 3.4% (n = 3). One patient succumbed to device-infection-related sepsis, another experienced refractory acute heart failure, and one patient passed away the day after the procedure due to an undetermined cause. Table 1 provides an overview of patient characteristics and indications for transvenous lead extraction (TLE). Detailed information on lead and device characteristics can be found in Table 2 and Table 3. Figure 4 and Figure 5 depict the non-powered equipment utilized for TLE.

Among the patients included in the study, 67% were males, with a mean age of 66.16 ± 16 years. The youngest patient was 18, while the oldest was 98. The average lead dwell time was 6.92 ± 4.47 years, ranging from a minimum of 1 year to a maximum of 26 years. More than half of the patients (52.8%) had leads older than 5 years, and 15.8% had leads older than 10. Of the participants, 73.9% required TLE due to infection, with 31.8% diagnosed with infective endocarditis or device-related sepsis and 42% presenting with pocket infection. Abandoned or dysfunctional leads were observed in 15.9% of cases, while venous occlusion was identified in 10.2%.

The transvenous approach was used in most patients (82.9%), with 13.6% requiring the femoral approach and 3.4% undergoing the jugular approach. Among patients diagnosed with infective endocarditis (31.8%), positive blood cultures were accompanied by documentation of intracardiac or lead vegetations using transthoracic echocardiography (TTE) and transesophageal echocardiography (TOE), without any other apparent cause for bacteremia. Vegetations were observed in 9% of patients. All vegetations measured <15 mm. One vegetation was adherent to the intra-cardiac RV coil, and most were attached to the tricuspid valve or the lead in the proximity of the tricuspid annulus. One was associated with severe tricuspid valve regurgitation and was later referred to cardiovascular surgery for extraction and valve repair after minimal efforts to perform TLE, which were unsuccessful in the end. Of note, fragments of vegetations in some cases were still observed on transthoracic echography after successful TLE.

In cases of pocket infection (42%), clinical presentations included device exteriorization (21.6%), a pocket fistula with continuous purulent drainage (45.9%), and definite signs of pocket inflammation such as redness, swelling, skin necrosis, pain, and firm adherence of the device to the skin (32.4%).

### Reimplantation Strategy

All patients had their indication for reimplantation re-evaluated. According to the updated guidelines, a reimplantation procedure was performed if the implant was still indicated. The reimplantation procedure in patients with previous pocket infections was performed on the contralateral side after as soon as a couple of days in pacemaker-dependent patients or after a minimum of two weeks in non-pacemaker-dependent patients.

In patients diagnosed with endocarditis and systemic infections, the reimplantation was performed only after the normalization of cardiac echography and inflammation markers, along with two negative blood cultures. Pacemaker-dependent patients were required to wait 30 days, and pacemaker-non-dependent patients were required to wait 90 days before they could undergo reimplantation. Temporary pacing via the right internal jugular vein was left in place during the waiting period for the pacemaker dependents. The contralateral implant was preferred in all patients except those with a waiting period of more than six months. With this strategy for pacemaker reimplantation, we report no re-infections in a 1-year follow-up.

## 4. Discussion

Our experience represents a single-center consecutive case series of 150 chronic endovascular and defibrillator lead extraction procedures performed over 4 years. Using only mechanical non-powered lead extraction tools and multiple venous route approaches when needed, we report a 93.3% complete lead removal success rate and a 99% clinical success rate. No major complications requiring medical intervention were noted. The 30-day mortality rate was 3.4%. Our success rates and 30-day mortality rate are like those in other TLE publications [40,41,42,43,44,45,46,47,48,49]. Patient characteristics, devices, and lead characteristics were in line with other studies published in the literature. The technique we used was first described by Bongiorni MG et al., reporting high success rates and safety using simple extraction tools [50]. The most common indication for TLE in our study was device-related infection (74%), like reports regarding TLE and device infections in the EU and US [51,52]. We started using this approach in 2018 and have become a referral center for TLE. Over the last five years, we have performed 100+ TLE procedures to extract over 180 pacing and defibrillator leads, and the numbers are increasing. The active fixation mechanism was the most frequent type of fixation found in targeted leads for removal (92.6%), as this type of fixation is the preferred type to be used for implants in our country. ICD leads (single and dual coil) were harder to extract because the coil is more prone to form strong intravascular and intracardiac adherence tissue; ICD leads represented 21.2% of all extracted leads. Extraction of coronary sinus (CS) leads can frequently be performed via simple traction, even if the lead dwell time is over 12 months. Nevertheless, dedicated extraction tools were also used for CS pacing leads. Ten percent of extracted leads were CS leads, and we only targeted the CS lead after removing the other leads from the right atrium and right ventricle, which probably helped simple traction success for CS leads.

In our experience, we have found that this technique is both feasible and highly effective, with an extremely low number of complications. None of the complications encountered were considered major or required further medical assistance. However, it is important to note that our study specifically focused on extracting very old leads, with a maximum lead age of 26 years. We followed strict procedures, including the transvenous lead extraction of a specific type of coronary sinus active fixation lead, Attain StarFix by Medtronic.

It should be emphasized that acquiring proficiency in this technique requires a learning curve, and we recommend that an experienced operator be involved to achieve optimal outcomes and minimize the risk of complications. The exact amount of training needed to become proficient in lead extraction is still to be determined. According to some guidance documents, a minimum requirement for training is the extraction of at least 40 leads in a minimum of 30 procedures. Additionally, a minimum of 15 procedures involving the removal of at least 20 leads annually is suggested to maintain competency [17,53,54].

As for reimplantation, other options such as leadless pacemakers and subcutaneous ICD may be considered, as suggested by various papers [44,45]. However, higher thresholds for pacing may occur in the setting of deploying a leadless pacemaker in the same RV area where previous leads have been removed [46]. In our center, leadless pacing for reimplantation was not used, and only one patient with an indication for ICD reimplantation had an S-ICD implanted (Figure 6).

In summary, while our experience has demonstrated the feasibility and effectiveness of this technique with minimal complications, the importance of proper training and experience cannot be understated. Continued research and guidelines will provide further insight into the training requirements for proficiency in lead extraction.

Regarding safety, it is crucial to emphasize the significance of conducting these procedures in a hybrid laboratory where a cardiovascular surgeon can quickly intervene in the event of a major complication. While cardiologists can manage certain cases of cardiac tamponade through draining the fluid, there are instances where immediate surgical intervention is necessary to minimize the risk of procedural death [55]. A large retrospective study by Kutarski et al., which included 2049 patients who underwent transvenous lead extraction (TLE), highlighted the importance of implementing a strategy that involves performing procedures in a cardiac surgery operating theater or hybrid room to prevent intraoperative fatalities [55]. It is important to note that, in the long run, performing TLE without on-site surgical backup should not be routinely conducted or advocated for to ensure optimal patient safety.

Clinical and laboratory indicators of device infections are often limited. Despite the importance of timely diagnosis for prognosis, many patients exhibit minimal or no symptoms of systemic inflammatory response. Additionally, blood cultures frequently yield negative results, even in cases of device-related endocarditis. The timely referral of patients could significantly reduce the 30-day mortality rate, as device-related septic complications, combined with patient comorbidities, substantially impact both short-term and long-term survival. Unfortunately, identifying predictors of mortality in our study was not feasible.

The demand for transvenous lead extraction (TLE) procedures is expected to grow exponentially due to the increasing number of cardiac implantable electronic device (CIED) implants and an aging population. However, the number of physicians trained in this field remains limited. There is a growing need for specialists to develop new techniques that enhance safety, reduce mortality rates, and achieve optimal outcomes.

The findings of our study may serve as an encouragement for more centers to adopt our strategy and safely perform TLE. The approach has demonstrated safety, effectiveness, and cost-effectiveness, which could ultimately benefit more patients who need lead extraction procedures.

### Study Limitations

This is a low-volume center retrospective experience of consecutive patients referred to TLE, and our findings are subject to bias and may be influenced by our management strategy.

One important limitation is the missing data on long-term follow-ups, such as 1-year survival rate and infection relapse in the long term. Another limitation is the fact that the largest number of patients referred to our center have already underwent a form of medical treatment, including antimicrobial therapy, which could have changed the status of bloodstream cultures before the procedure and may have modified the course of management through delaying the appropriate therapy, thus influencing the outcome.

We could not determine predictors of the 30-day mortality rate in our study. Further data and larger studies are needed to identify several risk factors for mortality and reinfection and evaluate long-term survival rates after TLE.

## 5. Conclusions

Our study has demonstrated that transvenous lead extraction is safe and highly effective and may be safely practiced in low-volume centers using non-powered extraction tools even without on-site surgical backup. The most common indication is a device-related infection, including pocket infection, endocarditis, and sepsis. Complications are rare in the presence of an experienced operator, and in most cases, they do not require further medical intervention and resolve spontaneously.

## Figures and Tables

**Figure 1 reports-06-00041-f001:**
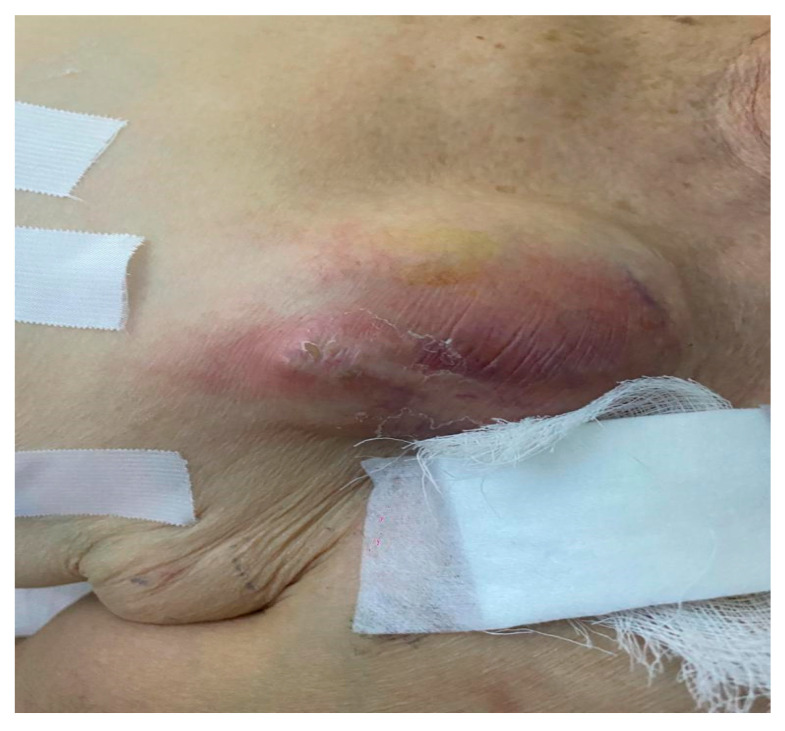
Fluctuant erythema and swelling of infected pocket.

**Figure 2 reports-06-00041-f002:**
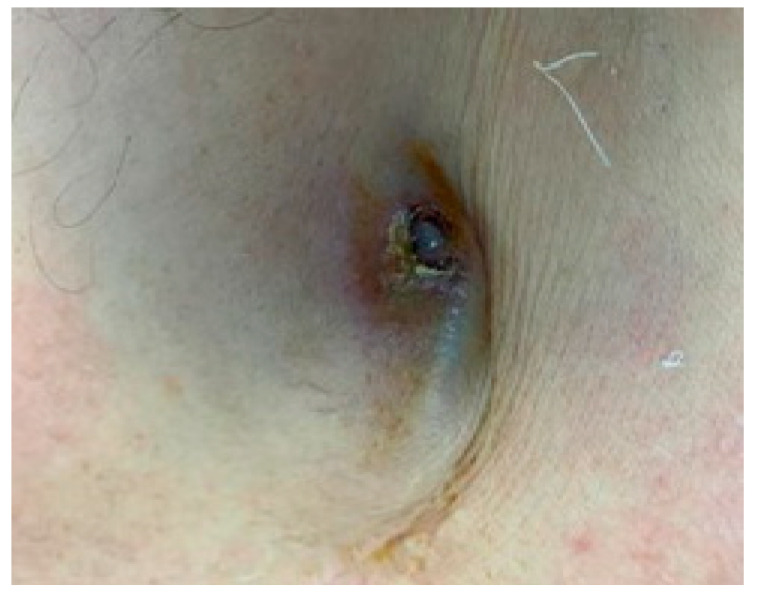
Skin adherence with erosion.

**Figure 3 reports-06-00041-f003:**
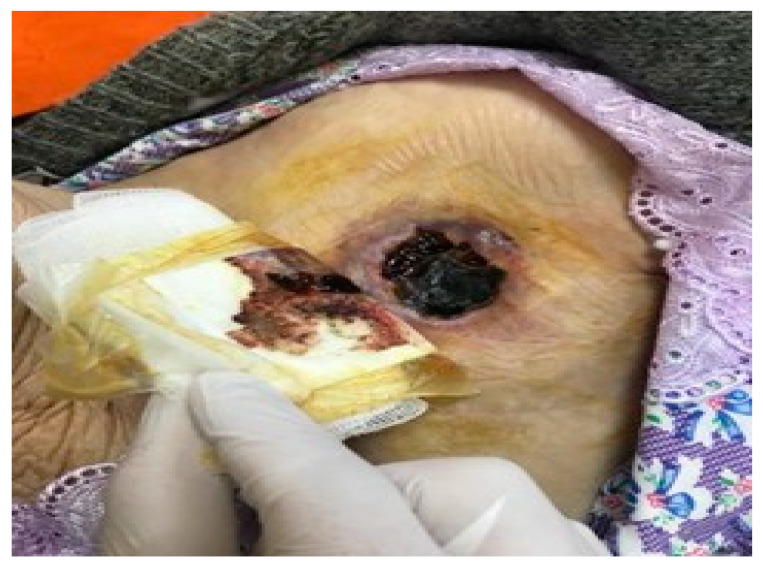
Skin necrosis.

**Figure 4 reports-06-00041-f004:**
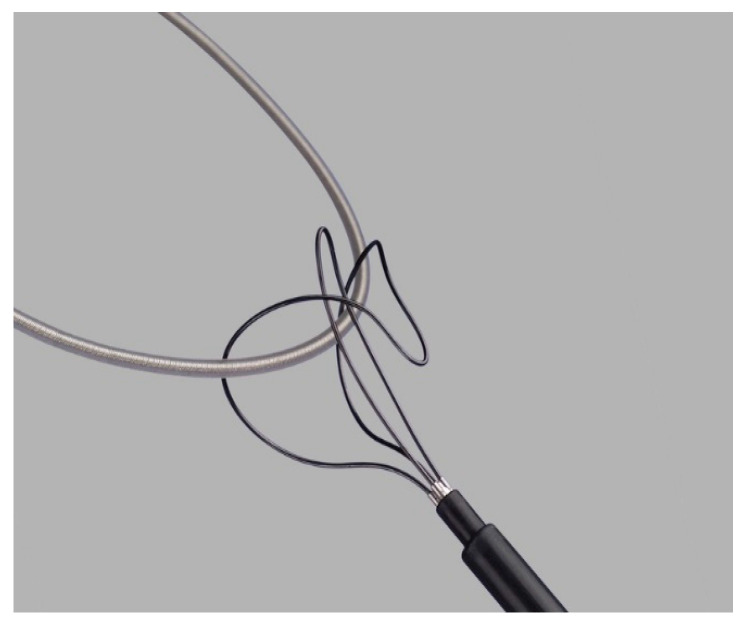
Needle eye snare (Cook Medical).

**Figure 5 reports-06-00041-f005:**
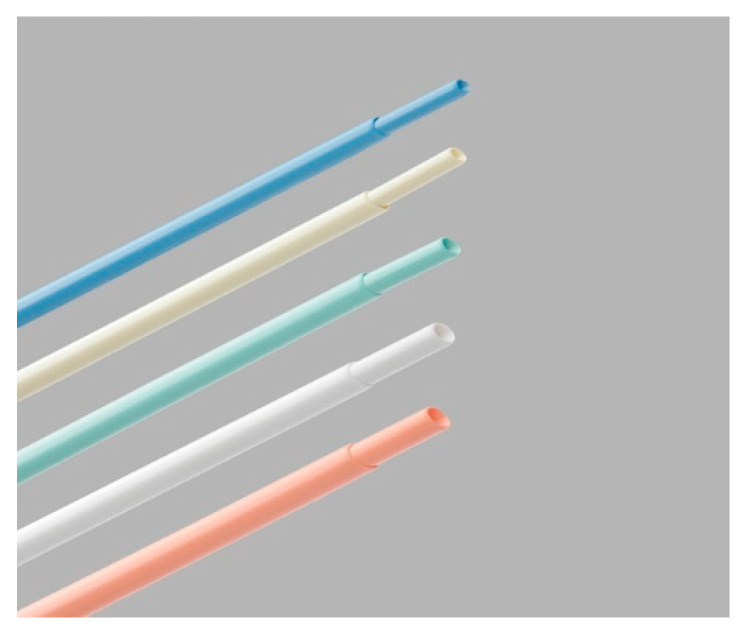
Byrd dilator sheath.

**Figure 6 reports-06-00041-f006:**
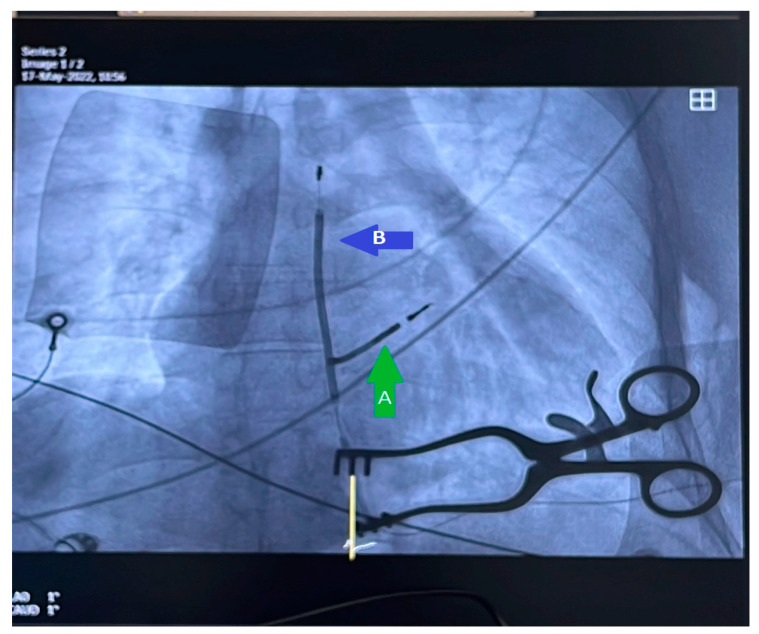
(A) Residual material after TLE procedure for removing single chamber single coil ICD in a 37-year-old patient with device-related endocarditis. Infection resolved after removing all materials except for the RV coil, which remained attached to the IV septum. One year later, the patient was implanted with an S-ICD; the new parasternal ICD coil can also be observed (B).

**Table 1 reports-06-00041-t001:** Patient characteristics.

Patient age, years, mean (standard deviation)	66.16 (16.00)	
Time since first implant, years mean (standard deviation)	6.92 (4.47) 0.477)	
Left ventricular EF, %, mean (standard deviation)	43.8% (14.06)	
Creatinine, mg/dL, mean (standard deviation)	1.00 (0.46)	
	Frequency = n	Percent%
Number of patients	88	100
Sex, male	59	67.0%
Co-morbidities		
HTN	55	65.5%
Ischaemic Cardiomyopathy	21	25%
Diabetes	23	27.4%
Chronic kidney disease	13	14.7%
Dyslipidemia	10	11.9%
Atrial fibrillation	36	42.9%
Anemia	52	63.4%
Indications for removal		
Infectious indication	65	74%
Endocarditis	28	31.8%
Pocket infection	37	42%
Non-infectious indication	23	26%
Venous occlusion	9	10.2%
Abandoned/disfunctional lead	14	15.9%

**Table 2 reports-06-00041-t002:** Extracted device type.

Device Type	Frequency = n	Percent%
VVI	12	13.6
DDD	32	36.4
CRT-P	7	8.0
SC-ICD	15	17.0
DC-ICD	7	8.0
CRT-D	15	17
Total	88	100.0

VVI: single-chamber pacemaker. DDD: dual-chamber pacemaker. SC: single chamber. DC: dual chamber. CRT: cardiac resynchronization therapy. ICD: implantable cardioverter defibrillator.

**Table 3 reports-06-00041-t003:** Characteristics of targeted leads.

Lead Type		
Lead age (mean, years)	6.92 ± 4.47 (1–26)	
>5 (years, leads n)	46	52.8%
>10 (years, leads n)	14	15.8%
Leads extracted per procedure (=n)		
Median	2 (1–4)	
1	31	38.2%
2	34	42%
3	14	17.3%
4	2	2.5%
Type of extracted leads	Frequency = n	Percent%
RA/RV pacing	102	68
ICD S-C	25	16.6
ICD D-C	7	4.6
CS pace	16	10.6
Type of fixation		
Active fixation	139	92.6
Passive fixation	11	7.3

S-C: single-coil. D-C: dual-coil. CS: coronary sinus. RA/RV: right atrium/ventricle. ICD: implantable cardioverter defibrillator.

## Data Availability

Not applicable.

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
