# Peer review of "Transvenous Lead Extraction in a European Low-Volume Center without On-Site Surgical Support"

_reports, 2023, doi:10.3390/reports6030041_

Round 1

Reviewer 1 Report

In this manuscript Dardri et al. present a single center retrospective experience of lead extraction experience framed without on-site surgical support at a single european center. A cohort of 88 consecutive patients were examined.  Although cohort size was relatively modest this study demonstrates the potential feasibility of lead extraction at experienced center without onsite surgical back up using a more novel technology/technique for TLE.  The authors should be congratulated on their excellent clinical outcomes with high success and low complications.  I believe this study is a good addition to the literature and offer the following constructive comments.

Comments:

1.  Introduction; second sentence:  The reviewer notices that the studies currently cited hare are quite dated as many are studies 15-20 years old.  Please consider adding the following contemporary references below:

Palmisano P, Guerra F, Dell'Era G, et al. Impact on All-Cause and Cardiovascular Mortality of Cardiac Implantable Electronic Device Complications: Results From the POINTED Registry. JACC Clin Electrophysiol. 2020 Apr;6(4):382-392. doi: 10.1016/j.jacep.2019.11.005.

Lemmermöhle, E., Lackermair, K., Klier, I. et al. Impact of lead detecting algorithms on inappropriate shocks in implantable cardioverter defibrillator lead failure: a single-center manufacturer-independent observational study. J Interv Card Electrophysiol 66, 1059–1061 (2023). https://doi.org/10.1007/s10840-022-01460-1

Kirkfeldt RE, Johansen JB, Nohr EA, Jørgensen OD, Nielsen JC. Complications after cardiac implantable electronic device implantations: an analysis of a complete, nationwide cohort in Denmark. Eur Heart J. 2014 May;35(18):1186-94. doi: 10.1093/eurheartj/eht511.

Witt, C.T., Ng Kam Chuen, M.J., Kronborg, M.B. et al. Non-infective left ventricular lead complications requiring re-intervention following cardiac resynchronization therapy: prevalence, causes and outcomes. J Interv Card Electrophysiol 63, 69–75 (2022). https://doi.org/10.1007/s10840-021-00947-7

Ganesan AN, Moore K, Horton D, et al. Complications of cardiac implantable electronic device placement in public and private hospitals. Intern Med J. 2020 Oct;50(10):1207-1216. doi: 10.1111/imj.14704. PMID: 31762133.

Merchant FM, Levy MR, Kelli HM, et al. Predictors of Long-Term Survival Following Transvenous Extraction of Defibrillator Leads. Pacing Clin Electrophysiol. 2015 Nov;38(11):1297-303. doi: 10.1111/pace.12733. 

2.  Introduction; lines 53-57:  References should be also added to support these statements regarding complication rates and types.  See studies below:

Sood N, Martin DT, Lampert R, Curtis JP, Parzynski C, Clancy J. Incidence and Predictors of Perioperative Complications With Transvenous Lead Extractions: Real-World Experience With National Cardiovascular Data Registry. Circ Arrhythm Electrophysiol. 2018 Feb;11(2):e004768. doi: 10.1161/CIRCEP.116.004768.

Khalil, M., Maqsood, M.H., Maraey, A. et al. Sex differences in outcomes of transvenous lead extraction: insights from National Readmission Database. J Interv Card Electrophysiol (2022). https://doi.org/10.1007/s10840-022-01438-z

Boarescu PM, Roşian AN, Roşian ŞH. Transvenous Lead Extraction Procedure-Indications, Methods, and Complications. Biomedicines. 2022 Nov 1;10(11):2780. doi: 10.3390/biomedicines10112780. 

3.  Introduction, Lines 57-58:  References should also definitely added to support authors' statement regarding success rates and for the sentence mentioning "few studies" studies that have examined safety and efficacy of TLE.  See following studies:

Pecha S, Ziegelhoeffer T, Yildirim Y, et al. Safety and efficacy of transvenous lead extraction of very old leads. Interact Cardiovasc Thorac Surg. 2021 Apr 8;32(3):402-407. doi: 10.1093/icvts/ivaa278. 

Stefańczyk P, Nowosielecka D, Polewczyk A, et al. Safety and Effectiveness of Transvenous Lead Extraction in Patients with Infected Cardiac Resynchronization Therapy Devices; Is It More Risky than Extraction of Other Systems? Int J Environ Res Public Health. 2022 May 10;19(10):5803. doi: 10.3390/ijerph19105803. 

Zsigmond, EJ., Miklos, M., Vida, A. et al. Reimplantation and long-term mortality after transvenous lead extraction in a high-risk, single-center cohort. J Interv Card Electrophysiol 66, 847–855 (2023). https://doi.org/10.1007/s10840-021-00974-4

Segreti L, Giannotti Santoro M, Di Cori A, Fiorentini F, Zucchelli G, Bernini G, De Lucia R, Viani S, Paperini L, Barletta V, Soldati E, Bongiorni MG. Safety and efficacy of transvenous mechanical lead extraction in patients with abandoned leads. Europace. 2020 Sep 1;22(9):1401-1408. doi: 10.1093/europace/euaa134.

Lee, S.Y., Allen, I.E., Diaz, C. et al. Efficacy and mortality of rotating sheaths versus laser sheaths for transvenous lead extraction: a meta-analysis. J Interv Card Electrophysiol 66, 1067–1075 (2023). https://doi.org/10.1007/s10840-021-01076-x

Sharma S, Ekeruo IA, Nand NP, Sundara Raman A, Zhang X, Reddy SK, Hariharan R. Safety and Efficacy of Transvenous Lead Extraction Utilizing the Evolution Mechanical Lead Extraction System: A Single-Center Experience. JACC Clin Electrophysiol. 2018 Feb;4(2):212-220. doi: 10.1016/j.jacep.2017.12.010.

4.  Methods:  Was this an observational study with patients prospectively enrolled but retrospectively analyzed given patients were consecutive or was this purely retrospective?

5.  Methods section; Diagnostic management – Would consider removing discussion of patient with medically-managed bacteria from aorto-femoral infection as this patient was not part of the cohort and perhaps irrelevant

6.  Methods Section; 2.4 TLE indication - This section includes a nice written explanation of the guidelines and different wounds type, but seems more appropriate to be placed in a review rather than a research study.  Perhaps this section should be removed or shortened.  If kept, if any images are come from another article and are not original images from the authors then they should be properly attributed to source.

7.  Method section; TLE procedure- This section refers to "pacemakers" but study appears to have included pacemaker, ICDs and CRT devices.  Please clarify.

8.  Results:  I believe there may be a typo for 30-day mortality.  3 of 88 patients died which would be 3.4% of cohort, not 2.64%  Please verify.

9.  Please consider adding a panels to Figure 7 showing fluoroscopy images using the Needle eye snare during a TLE case.

10.  Discussion; Page 12; Paragraph 4 - This is a good point that many device infections can be subtle and missed for various reasons. Also sometimes when patients are admitted to the hospital it may not always occur to the physician treatment team to think about the device as a potential infection source.  Would discuss paper by Paz et al regarding use of electronic medical alerts for patients with CIEDs admitted for sepsis.

Paz Rios, L.H., Minga, I., Gaznabi, S. et al. The impact of an electronic medical alert system for patients with cardiac implantable electronic devices and bacteremia. J Interv Card Electrophysiol 66, 525–529 (2023). https://doi.org/10.1007/s10840-022-01423-6

11.  Discussion; page 12; reimplant procedures; paragraph - Waiting 30 and 90 days for reimplantation is a challenging proposition for some patients, especially if pacemaker dependent or high risk of VAs. Should discuss that leadless pacemaker and subcutaneous ICD are now available options for patients who do not require CRT after TLE.  See suggested references below

Mitacchione G, Schiavone M, Gasperetti A, Arabia G, Breitenstein A, Cerini M, Palmisano P, Montemerlo E, Ziacchi M, et al. Outcomes of leadless pacemaker implantation following transvenous lead extraction in high-volume referral centers: Real-world data from a large international registry. Heart Rhythm. 2023 Mar;20(3):395-404. doi: 10.1016/j.hrthm.2022.12.002.

Breeman, K.T.N., Beurskens, N.E.G., Driessen, A.H.G. et al. Timing and mid-term outcomes of using leadless pacemakers as replacement for infected cardiac implantable electronic devices. J Interv Card Electrophysiol (2022). https://doi.org/10.1007/s10840-022-01457-w

Beccarino NJ, Choi EY, Liu B, et al. Concomitant leadless pacing in pacemaker-dependent patients undergoing transvenous lead extraction for active infection: Mid-term follow-up. Heart Rhythm. 2023 Jun;20(6):853-860. doi: 10.1016/j.hrthm.2023.02.003.

Giacomin, E., Falzone, P.V., Dall’Aglio, P.B. et al. Subcutaneous implantable cardioverter defibrillator after transvenous lead extraction: safety, efficacy and outcome. J Interv Card Electrophysiol (2022). https://doi.org/10.1007/s10840-022-01293-y

Russo V, Viani S, Migliore F, Nigro G, Biffi M, Tola G, Bisignani G, Dello Russo A, Sartori P, Rordorf R, Ottaviano L, Perego GB, Checchi L, Segreti L, Bertaglia E, Lovecchio M, Valsecchi S, Bongiorni MG. Lead Abandonment and Subcutaneous Implantable Cardioverter-Defibrillator (S-ICD) Implantation in a Cohort of Patients With ICD Lead Malfunction. Front Cardiovasc Med. 2021 Jul 27;8:692943. doi: 10.3389/fcvm.2021.692943. 

Author Response

In this manuscript Dardri et al. present a single center retrospective experience of lead extraction experience framed without on-site surgical support at a single european center. A cohort of 88 consecutive patients were examined.  Although cohort size was relatively modest this study demonstrates the potential feasibility of lead extraction at experienced center without onsite surgical back up using a more novel technology/technique for TLE.  The authors should be congratulated on their excellent clinical outcomes with high success and low complications.  I believe this study is a good addition to the literature and offer the following constructive comments.

Comments:

  1. Introduction; second sentence:  The reviewer notices that the studies currently cited here are quite dated as many are studies 15-20 years old.  Please consider adding the following contemporary references below:

We have added the suggested more recent references.

Palmisano P, Guerra F, Dell'Era G, et al. Impact on All-Cause and Cardiovascular Mortality of Cardiac Implantable Electronic Device Complications: Results From the POINTED Registry. JACC Clin Electrophysiol. 2020 Apr;6(4):382-392. doi: 10.1016/j.jacep.2019.11.005.

Lemmermöhle, E., Lackermair, K., Klier, I. et al. Impact of lead detecting algorithms on inappropriate shocks in implantable cardioverter defibrillator lead failure: a single-center manufacturer-independent observational study. J Interv Card Electrophysiol 66, 1059–1061 (2023). https://doi.org/10.1007/s10840-022-01460-1

Kirkfeldt RE, Johansen JB, Nohr EA, Jørgensen OD, Nielsen JC. Complications after cardiac implantable electronic device implantations: an analysis of a complete, nationwide cohort in Denmark. Eur Heart J. 2014 May;35(18):1186-94. doi: 10.1093/eurheartj/eht511.

Witt, C.T., Ng Kam Chuen, M.J., Kronborg, M.B. et al. Non-infective left ventricular lead complications requiring re-intervention following cardiac resynchronization therapy: prevalence, causes and outcomes. J Interv Card Electrophysiol 63, 69–75 (2022). https://doi.org/10.1007/s10840-021-00947-7

Ganesan AN, Moore K, Horton D, et al. Complications of cardiac implantable electronic device placement in public and private hospitals. Intern Med J. 2020 Oct;50(10):1207-1216. doi: 10.1111/imj.14704. PMID: 31762133.

Merchant FM, Levy MR, Kelli HM, et al. Predictors of Long-Term Survival Following Transvenous Extraction of Defibrillator Leads. Pacing Clin Electrophysiol. 2015 Nov;38(11):1297-303. doi: 10.1111/pace.12733. 

  1.  Introduction; lines 53-57:  References should be also added to support these statements regarding complication rates and types.  See studies below:

We have added suggested references specifically to highlight the importance of immediate surgical backup.

Sood N, Martin DT, Lampert R, Curtis JP, Parzynski C, Clancy J. Incidence and Predictors of Perioperative Complications With Transvenous Lead Extractions: Real-World Experience With National Cardiovascular Data Registry. Circ Arrhythm Electrophysiol. 2018 Feb;11(2):e004768. doi: 10.1161/CIRCEP.116.004768.

Khalil, M., Maqsood, M.H., Maraey, A. et al. Sex differences in outcomes of transvenous lead extraction: insights from National Readmission Database. J Interv Card Electrophysiol (2022). https://doi.org/10.1007/s10840-022-01438-z

Boarescu PM, Roşian AN, Roşian ŞH. Transvenous Lead Extraction Procedure-Indications, Methods, and Complications. Biomedicines. 2022 Nov 1;10(11):2780. doi: 10.3390/biomedicines10112780. 

  1.  Introduction, Lines 57-58:  References should also definitely added to support authors' statement regarding success rates and for the sentence mentioning "few studies" studies that have examined safety and efficacy of TLE.  See following studies:

We have added some of the suggested papers.

Pecha S, Ziegelhoeffer T, Yildirim Y, et al. Safety and efficacy of transvenous lead extraction of very old leads. Interact Cardiovasc Thorac Surg. 2021 Apr 8;32(3):402-407. doi: 10.1093/icvts/ivaa278. 

Stefańczyk P, Nowosielecka D, Polewczyk A, et al. Safety and Effectiveness of Transvenous Lead Extraction in Patients with Infected Cardiac Resynchronization Therapy Devices; Is It More Risky than Extraction of Other Systems? Int J Environ Res Public Health. 2022 May 10;19(10):5803. doi: 10.3390/ijerph19105803. 

Zsigmond, EJ., Miklos, M., Vida, A. et al. Reimplantation and long-term mortality after transvenous lead extraction in a high-risk, single-center cohort. J Interv Card Electrophysiol 66, 847–855 (2023). https://doi.org/10.1007/s10840-021-00974-4

Segreti L, Giannotti Santoro M, Di Cori A, Fiorentini F, Zucchelli G, Bernini G, De Lucia R, Viani S, Paperini L, Barletta V, Soldati E, Bongiorni MG. Safety and efficacy of transvenous mechanical lead extraction in patients with abandoned leads. Europace. 2020 Sep 1;22(9):1401-1408. doi: 10.1093/europace/euaa134.

Lee, S.Y., Allen, I.E., Diaz, C. et al. Efficacy and mortality of rotating sheaths versus laser sheaths for transvenous lead extraction: a meta-analysis. J Interv Card Electrophysiol 66, 1067–1075 (2023). https://doi.org/10.1007/s10840-021-01076-x

Sharma S, Ekeruo IA, Nand NP, Sundara Raman A, Zhang X, Reddy SK, Hariharan R. Safety and Efficacy of Transvenous Lead Extraction Utilizing the Evolution Mechanical Lead Extraction System: A Single-Center Experience. JACC Clin Electrophysiol. 2018 Feb;4(2):212-220. doi: 10.1016/j.jacep.2017.12.010.

  1.  Methods:  Was this an observational study with patients prospectively enrolled but retrospectively analyzed given patients were consecutive or was this purely retrospective?

This study was purely retrospective, as more and more procedures were performed with high success rates, we decided to start analyzing the outcomes.

  1.  Methods section; Diagnostic management – Would consider removing discussion of patient with medically-managed bacteria from aorto-femoral infection as this patient was not part of the cohort and perhaps irrelevant

We have removed the above-mentioned patient but left in place the comments about our approach for the use of PET/CT, as it’s mentioned as a potentially useful diagnostic tool.

  1.  Methods Section; 2.4 TLE indication - This section includes a nice written explanation of the guidelines and different wounds type, but seems more appropriate to be placed in a review rather than a research study.  Perhaps this section should be removed or shortened.  If kept, if any images are come from another article and are not original images from the authors then they should be properly attributed to source.

We have shortened the section by removing table 1. We have kept definitions and figures. All figures (1-5) are from our center.

  1.  Method section; TLE procedure- This section refers to "pacemakers" but study appears to have included pacemaker, ICDs and CRT devices.  Please clarify.

Indeed there were all types of devices (PM, ICD and CRT). We replaced the word pacemaker with device, to eliminate the confusion.

  1.  Results:  I believe there may be a typo for 30-day mortality.  3 of 88 patients died which would be 3.4% of cohort, not 2.64%  Please verify.

Sorry for the typo, we have revised and corrected it wherever it appears in text.

  1.  Please consider adding a panels to Figure 7 showing fluoroscopy images using the Needle eye snare during a TLE case.

Unfortunately, the fluoroscopy system is temporarily under service/revision, and we cannot access fluoroscopy recordings of procedures.

  1.  Discussion; Page 12; Paragraph 4 - This is a good point that many device infections can be subtle and missed for various reasons. Also sometimes when patients are admitted to the hospital it may not always occur to the physician treatment team to think about the device as a potential infection source.  Would discuss paper by Paz et al regarding use of electronic medical alerts for patients with CIEDs admitted for sepsis.

Paz Rios, L.H., Minga, I., Gaznabi, S. et al. The impact of an electronic medical alert system for patients with cardiac implantable electronic devices and bacteremia. J Interv Card Electrophysiol 66, 525–529 (2023). https://doi.org/10.1007/s10840-022-01423-6

  1.  Discussion; page 12; reimplant procedures; paragraph - Waiting 30 and 90 days for reimplantation is a challenging proposition for some patients, especially if pacemaker dependent or high risk of VAs. Should discuss that leadless pacemaker and subcutaneous ICD are now available options for patients who do not require CRT after TLE.  See suggested references below

We have commented on the leadless pacing and S-ICD options and added some references in the discussion section. Thanks for the reminder about these options. We have used an S-ICD for reimplanting a young 37 Y.O patient with previous device-related endocarditis on an endocardial RV lead.  

Mitacchione G, Schiavone M, Gasperetti A, Arabia G, Breitenstein A, Cerini M, Palmisano P, Montemerlo E, Ziacchi M, et al. Outcomes of leadless pacemaker implantation following transvenous lead extraction in high-volume referral centers: Real-world data from a large international registry. Heart Rhythm. 2023 Mar;20(3):395-404. doi: 10.1016/j.hrthm.2022.12.002.

Breeman, K.T.N., Beurskens, N.E.G., Driessen, A.H.G. et al. Timing and mid-term outcomes of using leadless pacemakers as replacement for infected cardiac implantable electronic devices. J Interv Card Electrophysiol (2022). https://doi.org/10.1007/s10840-022-01457-w

Beccarino NJ, Choi EY, Liu B, et al. Concomitant leadless pacing in pacemaker-dependent patients undergoing transvenous lead extraction for active infection: Mid-term follow-up. Heart Rhythm. 2023 Jun;20(6):853-860. doi: 10.1016/j.hrthm.2023.02.003.

Giacomin, E., Falzone, P.V., Dall’Aglio, P.B. et al. Subcutaneous implantable cardioverter defibrillator after transvenous lead extraction: safety, efficacy and outcome. J Interv Card Electrophysiol (2022). https://doi.org/10.1007/s10840-022-01293-y

Russo V, Viani S, Migliore F, Nigro G, Biffi M, Tola G, Bisignani G, Dello Russo A, Sartori P, Rordorf R, Ottaviano L, Perego GB, Checchi L, Segreti L, Bertaglia E, Lovecchio M, Valsecchi S, Bongiorni MG. Lead Abandonment and Subcutaneous Implantable Cardioverter-Defibrillator (S-ICD) Implantation in a Cohort of Patients With ICD Lead Malfunction. Front Cardiovasc Med. 2021 Jul 27;8:692943. doi: 10.3389/fcvm.2021.692943. 

Reviewer 2 Report

This manuscript, submitted by Dardari M, et al, is a retrospective analysis to evaluate the safety and effectiveness of Bongiorni’s technique with no on-site surgical backup in a low-volume center. By using only mechanical non-powered lead-extraction tools and multiple venous route approaches, the authors achieved 93.3% complete lead removal success rate and a 99% clinical success. Furthermore, no major complications requiring medical intervention occurred and the 30-day mortality rate was low. On the other hand, the following points need to be revised appropriately for the acceptance of this paper.

Specific comments

#1. The structure of results is complicated and hard to read. It needs to be revised, such as breaking it into paragraphs instead of bullets.

#2. Figures 1 through 5 need to be reduced because there are too many figures. Please reconsider.

#3. Please delete Figure 7.

#4. Please write the section of re-implant procedure in the text of the section of result.

Author Response

This manuscript, submitted by Dardari M, et al, is a retrospective analysis to evaluate the safety and effectiveness of Bongiorni’s technique with no on-site surgical backup in a low-volume center. By using only mechanical non-powered lead-extraction tools and multiple venous route approaches, the authors achieved 93.3% complete lead removal success rate and a 99% clinical success. Furthermore, no major complications requiring medical intervention occurred and the 30-day mortality rate was low. On the other hand, the following points need to be revised appropriately for the acceptance of this paper.

Specific comments

#1. The structure of results is complicated and hard to read. It needs to be revised, such as breaking it into paragraphs instead of bullets.

We have rearranged the results section and broken it into paragraphs for ease of following.

#2. Figures 1 through 5 need to be reduced because there are too many figures. Please reconsider.

We have reduced the number of figures.

#3. Please delete Figure 7.

We believe it’s representative of the tool used.

#4. Please write the section of re-implant procedure in the text of the section of result.

We have moved it to the results section. Thanks for the tip.

Reviewer 3 Report

This is a retrospective analysis of a single-center case series of 88 patients with cardiac implantable electronic devices (CIEDs) undergoing transvenous lead extraction (TLE) procedures between October 2018 and July 2022. The aim of the study is to evaluate the safety and efficacy of transvenous lead extraction (TLE) using non-powered extraction tools and in the absence of an on-site cardiac surgery stand-by.

There are some major issues:

1.    The small number of patients does not allow to provide any indication that could contrast existing guidelines, which recommend management of CIED infection patients in a referral center with immediate surgical facilities. Indeed, the overall risk of serious complications after TLE is <2% (as per the ELECTRA registry), such that a study on 88 cases has no power to detect complications. Of note, 74% of patients included in this study underwent extraction due to device related infections.

Thus, the study could be reconsidered if proposed as a proof-of-concept or as an hypothesis-generating study that could pave the way for a randomized study with risk stratification (see below).

2.    Among patients with CIED infections, 31% had endocarditis. No data about the vegetation size and tricuspid valve involvement are given in the manuscript, although these parameters are known to significantly influence the outcome of both the disease and the procedure.

3.    In Table 4, the number of leads extracted per procedure should be clarified. The number of leads may significantly affect procedural success.

4.    Authors should clarify in the methods section how numerical data are presented (mean and standard deviation or median and interquartile range?). Therefore, manuscript and tables should be extensively revised accordingly.

5.    Please revise Caption of Tables 3 and 4. Abbreviations should be added as footnotes.

6.    The paragraph 7 should be moved before the discussion section.

Authors should put their observations into context of existing literature; specifically, should try to provide speculative evidence for risk conditions that would suggest surgical stand-by or, in contrast, low risk scenarios where this would be dispensable. This is a very important point as the need for surgical stand-by limits significantly the number of clinical sites performing TLE and there could well be cases where surgeon could be unnecessary.

Author Response

This is a retrospective analysis of a single-center case series of 88 patients with cardiac implantable electronic devices (CIEDs) undergoing transvenous lead extraction (TLE) procedures between October 2018 and July 2022. The aim of the study is to evaluate the safety and efficacy of transvenous lead extraction (TLE) using non-powered extraction tools and in the absence of an on-site cardiac surgery stand-by.

There are some major issues:

  1. The small number of patients does not allow to provide any indication that could contrast existing guidelines, which recommend management of CIED infection patients in a referral center with immediate surgical facilities. Indeed, the overall risk of serious complications after TLE is <2% (as per the ELECTRA registry), such that a study on 88 cases has no power to detect complications. Of note, 74% of patients included in this study underwent extraction due to device related infections. 

Thus, the study could be reconsidered if proposed as a proof-of-concept or as an hypothesis-generating study that could pave the way for a randomized study with risk stratification (see below).

Thanks for the suggestion. However, our goal was never to contrast the guidelines, we have underlined in many instances the importance of on-site surgical back-up and cited papers addressing this issue. We are not trying to encourage performing TLE with the lack of surgical back-up, we had on-call availability for surgical back-up. As we mentioned in the discussion section, our goal is to implement a strategy where all TLE procedures are performed in a hybrid operating room with readily available cardiovascular surgeon.

  1. Among patients with CIED infections, 31% had endocarditis. No data about the vegetation size and tricuspid valve involvement are given in the manuscript, although these parameters are known to significantly influence the outcome of both the disease and the procedure.

We have added data about the vegetations found.

  1. In Table 4, the number of leads extracted per procedureshould be clarified. The number of leads may significantly affect procedural success.

We have clarified the numer of leads extracted per procedure.

  1. Authors should clarify in the methods section how numerical data are presented (mean and standard deviation or median and interquartile range?). Therefore, manuscript and tables should be extensively revised accordingly. 

We have revised.

  1. Please revise Caption of Tables 3 and 4. Abbreviations should be added as footnotes.

We have moved the abbreviations under the tables.

  1. The paragraph 7 should be moved before the discussion section.

 We have moved it to the results section. Thanks for the tip.

Authors should put their observations into context of existing literature; specifically, should try to provide speculative evidence for risk conditions that would suggest surgical stand-by or, in contrast, low risk scenarios where this would be dispensable. This is a very important point as the need for surgical stand-by limits significantly the number of clinical sites performing TLE and there could well be cases where surgeon could be unnecessary.

Round 2

Reviewer 1 Report

Thank you.  No further comments

Reviewer 2 Report

The problems pointed out last time have been properly revised, and there are no additional comments.

Reviewer 3 Report

Authors have answered most of my comments.

They did not follow the suggestion to propose features that can identify subjects who can be safely opertated without surgical stand by.